# Light Hole Excitons in Strain-Coupled Bilayer Quantum Dots with Small Fine-Structure Splitting

Xiangjun Shang [1,2,3], Hanqing Liu [1,2,3], Xiangbin Su [1,2,3], Shulun Li [1,2,3], Huiming Hao [1,2,3], Deyan Dai [1,2,3], Zesheng Chen [1,2,3], Haiqiao Ni [1,2,3] and Zhichuan Niu [1,2,3,*]

1   State Key Laboratory for Superlattices and Microstructures, Institute of Semiconductors, Chinese Academy of Sciences, Beijing 100083, China
2   Center of Materials Science and Optoelectronics Engineering, University of Chinese Academy of Sciences, Beijing 100049, China
3   Beijing Academy of Quantum Information Sciences, Beijing 100193, China
*   Correspondence: zcniu@semi.ac.cn

**Abstract:** In this work, we measure polarization-resolved photoluminescence spectra from excitonic complexes in tens of single InAs/GaAs quantum dots (QDs) at the telecom O-band with strain-coupled bilayer structure. QDs often show fine-structure splitting (FSS) ~100 µeV in uniform anisotropy and valence-band mixing of heavy holes (HH) and light holes (LH); the biaxial strain also induces LH excitons with small FSS (especially XX, <5 µeV, 70% of QDs); delocalized LH reduces the Coulomb interaction between holes $V_{hh}$ and enhances population on LH excitons XX, $XX_{11}$, $X_{11}^{+}$ and $XX_{21}^{+}$.

**Keywords:** strain-coupled bilayer QDs; fine-structure splitting; valence-band mixing; light hole





## 1. Introduction

Epitaxial semiconductor quantum dots (QDs) can be uniformly fabricated in-plane with well-controlled growth parameters and optical performance, showing advantages in laser diodes and photodetectors [1–3]. In the quantum field, among various types of optically excited quantum emitters, single QDs in areal density $<1 \times 10^9$ cm$^{-2}$ cooled at temperature (T) <60 K show multiple exciton states with discrete spectral lines; its microcavity integration for enhancement and coupling in single-mode fiber (with little chromatic or polarization mode dispersion) and photonic integrated circuit (PIC) [4–8] pave the way for information processing; telecom-band single photons enable 100-km fiber transmission and coupling in silicon-based PIC on mature silicon-on-insulator substrate. For QDs grown on GaAs substrate compatible with lattice-matched GaAs/Al(Ga)As distributed Bragg reflector (DBR) cavity to enhance vertical extraction, droplet and Stranski–Krastanov (S-K) epitaxy have proved single QDs in a wide wavelength (λ) range, 0.7~1.6 µm [9–12]. A delicate design of QD structures (e.g., QD size by deposition amount control [13], substrate orientation [14]/misorientation [15], strain-reducing layer (SRL) as cap or buffer [10,11,15,16], modulated doping [17–19] and tunneling [19]) is crucial for QD optical performance. Strain-coupled bilayer InAs QDs with the lower layer to create strain nucleation sites and the upper layer to form strain-relaxed larger QDs aligned, capped by InGaAs SRL, have proved single QDs at λ ~1.3 µm [10]. Similar to its band tuning for high mobility [20], the strain engineering localized in the bilayer provides a tuning of exciton properties. Although photoluminescence (PL) study of QD ensembles has been fulfilled [21], a micro-PL (µPL) study of single QDs in such structure offers a direct probe to understand the strain effect on the QD electronic structure as referred to its morphology characterization and makes QD structure optimization more active. In this work, by extensive µPL study of many such kind of QDs in one sample in various exciton energies and configurations, it is found that the biaxial tensile strain in GaAs spacer below the upper QD induces a uniform QD anisotropy

with fine-structure splitting (FSS) ~100 μeV and light hole (LH) level with valence-band mixing (VBM) to heavy hole (HH) by tunneling (energy-dependent, higher-energy branch in a broader linewidth) as the degree of linear polarization (DLP) reflects [22]. LH excitons show that FSS is much reduced and becomes dominant when LHs delocalize, in particular biexciton $XX_{lh}$, with FSS as small as 0~5 μeV (70% of QDs), which is a little saturated under high pump when many LH excitons from the s-s or p-s transition in $C_{3v}$ features are built by delocalized LHs, unlike a weak LH exciton in QD with a buried strain layer [23]. The delocalized LHs also reduce the FSS of HH excitons. Charge defects build an electric field to vary the FSS oscillation phase and reduce exciton formation, with a clear spectral shoulder originating from the phonon. While still posing challenges (e.g., careful control of strain and interface defect to enhance QD brightness, optimizing the microcavity to enhance extraction, adding a dielectric mask to avoid surface adsorption or scratch, etching mesa to reduce the bulk strain), the prospect of this hybrid quantum structure as a QD molecule [21,24] is encouraged.

## 2. Materials and Methods

The bilayer InAs QDs are grown in molecular beam epitaxy on semi-insulating GaAs (001) substrate with a gradient indium flux: ultralow deposition rate and higher T for the lower seed layer; higher deposition rate and lower T for the upper layer capped by a 5 nm $In_{0.15}Ga_{0.85}As$ SRL; a 8 nm GaAs layer between them acts as both cap and space with strain field to form aligned QDs; a planar GaAs/$Al_{0.9}Ga_{0.1}As$ DBR cavity at λ ~1.3 μm (Q ~400) is integrated to enhance light extraction in this λ. For detail of the structure and growth, see [10,25]. To reflect QD size distribution, a test sample with the bottom DBR but no cavity filtering is grown to characterize the full spectrum of QDs and the atomic force microscope (AFM) morphology of uncapped QDs. Single-QD μPL spectrum is measured by a fiber-based confocal microscope spectrograph composed of an NA ~0.7 objective for collection with laser spot focused on the sample in diameter ~2 μm, a collimator coupling in single-mode fiber and a 0.5 m-long grating spectrometer equipped with a liquid nitrogen-cooled InGaAs linear array detector (Princeton Instruments). The sample is cooled at T ~5 K in a vibration-free helium-flow cryostat [26] and continuous-wave (cw) excited by a λ = 632.8 nm HeNe laser with the maximal power at the spot $P_0$ ~50 μW attenuated by a gradually varied neutral density filter for power-dependent measurement. A rotating half-wave plane (HWP) is inserted in front of a linear polarizer inline before fiber collection to filter out the distinct fine-structure component to observe energy oscillation beyond the spectrometer resolution, with peak energy of an exciton line in time-integrated PL spectra deduced from line fitting [17]. In the same QD, biexciton XX usually exhibits FSS oscillation in the same size but opposite sign to exciton X to emit polarization-correlated photon pairs. The LH–HH VBM [22] causes DLP, defined by $(I_x - I_y)/(I_x + I_y)$, where x and y mean two orthogonal axes, [1–10] and [110]. The slight offset in $LH_X$ and $LH_Y$ is reflected from FSS oscillation too. To understand LH, QD band structure is simulated by *Nextnano* in 8-band k.p theory with elastic strain minimized. In fact, the bulk strain in a planar sample is significant to affect QD performance during cryogen cycles. Etching mesa will reduce bulk strain for a better control of QD condition for quantum emission.

## 3. Results

### 3.1. PL Spectra and AFM Images of Bilayer QDs

The bilayer QD growth is found to be more sensitive on indium surface migration driven by strain sites and growth T. In the test sample with capped and uncapped QDs, Figure 1 presents their AFM morphology and PL spectra. The statistics on QD AFM height show correspondence to their PL spectra: larger indium coverage in region 1 leads to individual QDs at λ ~1.32 μm at a height of ≥13 nm (28/4 μm$^{-2}$), while lower coverage in region 2 leads to fewer QDs at a height of ≥13 nm (3/4 μm$^{-2}$) and a strong profile at λ ~1.05 μm from dense QDs at a height of 9~10 nm. For single-layer QD growth with indium coverage increasing, the third critical coverage forms large QDs at λ ~1.1 μm and

1.2 μm with a quantized height of ~13 nm and 15 nm after the first one to form single QDs at λ ~0.88 μm at a height of 1~2 nm and the second one to form single QDs at λ ~0.91 μm at a height of 7~8 nm [13] after 2D–3D transition [27]. Here, the strain relaxation and possible interlayer level coupling form single QDs at a height of ~13 nm with redshift λ ~1.3 μm. The excited states [28] also appear as the arrows indicate. Their slight redshift in region 2 reflects sufficient migration to form larger QDs due to a higher T; in the wafer center (region 3) with much higher T and lower coverage, the λ of lower-density large QDs even extends to 1.36 μm. In AFM images, compared to region 1, region 2 with enhanced migration shows large QDs in a larger base and more small QDs at a height of 5~7 nm. For bright single-photon emission at λ ~1.3 μm, the lower T in region 1 to form defectless high-aspect-ratio QDs with exciton lines appearing in spectral profile is desired, i.e., purely strain-driven (instead of T-driven) surface migration. Single QDs can be filtered (spatially isolated) by DBR cavity (e.g., pillar) [26,29], in addition to good control of indium coverage and strain site (see the spectrum in black curve, i.e., another point in region 1, with individual QDs at λ ~1.3 μm in sharp exciton lines). Compared to the narrow growth T window for the bilayer single QDs at λ ~1.3 μm, the case is different for single QDs at λ <1.1 μm, with the second critical coverage where a broad growth T exists (both region 1 and region 2 with various growth T show exciton lines in the spectral profile). QDs in the larger base likely have more interface defects to build a multielectric field to reduce exciton formation and show a smooth spectral profile from phonon broadening, with the excited states relatively higher populated. Since QD emission redshifts are ~100 nm at room T, the bilayer structure is also valuable for growth of QD ensemble at λ ~1.45 μm [12].

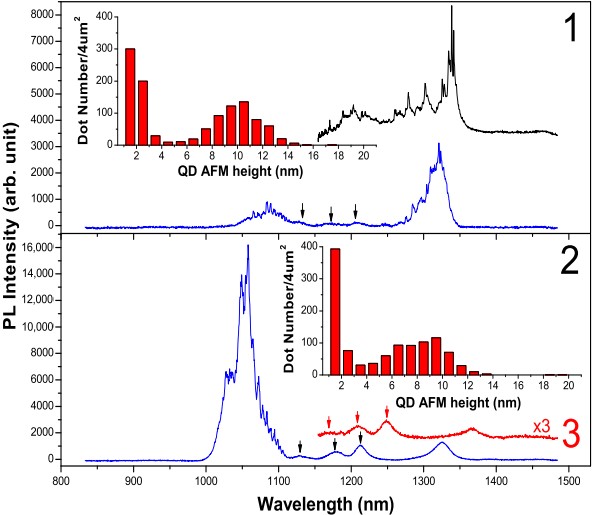
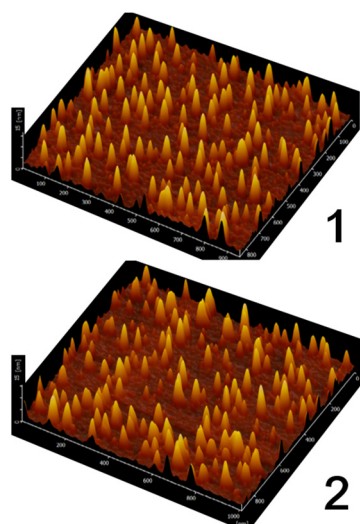

**Figure 1.** (**left**): μPL spectra of bilayer QDs: region 1 shows individual QDs at λ ~1.32 μm with exciton lines, two points, in blue and black curves; region 2 shows individual QDs at λ ~1.32 μm in lower aspect ratio and smooth spectral profile, a strong profile at λ ~1.05 μm from small QDs; region 3 shows extended λ; (**right**): AFM images; (**inset**): statistics on QD AFM height.

### 3.2. Band Structure of Bilayer Single QD

The simulation result is presented in Figure 2. There are the electron, HH and LH ground states E1, LH1 and HH1; with proper QD size, their energy offsets agree with the emission λ ~1.3 μm. HH in p-type as simulated is localized in the upper QD, while LH in s-type is confined in the thin GaAs spacer below it where the biaxial tensile strain lifts the LH band. The biaxial strain mainly affects the valence band, enlarging HH anisotropy and FSS and building LH excitons in $C_{3v}$ symmetry with small eh exchange (overlap) and FSS. An LH–HH mixing state as presented is responsible for DLP ~30% in polarization-resolved spectra below. The interlayer mixing is via LH tunneling (HH hardly tunnel), sensitive on its energy (i.e., spin states or polarization) and showing different spectral linewidths.

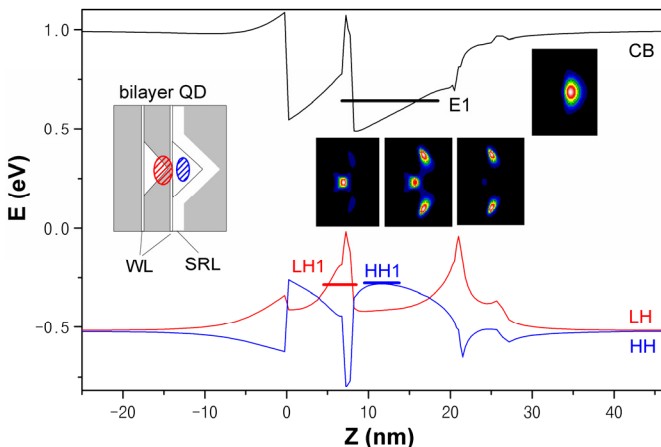

**Figure 2.** Band structure of bilayer QD as schematized (WL: wetting layer). Insets: ground-state wavefunctions.

### 3.3. QDs with Strain-Induced Large FSS

Figure 3 presents polarization-resolved PL spectra of six single-QDs in a planar DBR cavity in orthogonal axes, red [1–10] (horizontal, H) and black [110] (vertical, V). There are LHs and HHs to form HH excitons and LH excitons as marked, especially $XX_{lh}$. FSS > 100 μeV in HH excitons X and XX with opposite signs reflect a large anisotropy in the bilayer QDs compared to single-layer QDs at λ ~0.9 μm (FSS ~30 μeV typically). The uniform FSS oscillation phases, lower-energy branch along 90° [110] V for X and 0° [1–10] H for XX, reflect a uniform QD elongation (or anisotropy) along [110] V formed by the biaxial strain. The dominant X shows different spectral linewidths in the two polarizations: higher-energy branch [1–10] H with more LH interlayer tunneling for mixing in a broader linewidth. XX with the two branches $LH_X$ and $LH_X$ filled shows the same linewidth in both polarizations. In the lower-energy side of X, there is an exciton in thinner linewidth with FSS ~0, attributed to the negative trion $X^-$. For HH and LH biexcitons XX and $XX_{lh}$, there is an $XX_1\bar{1}$ ($2e1h_11h_2$) exciton in the lower-energy side with nearly equal energy offset. In QD2, QD4 and QD5, the $XX_1\bar{1}$ for HH ($C_{2v}$, LH $h_2$) shows negligible splitting, while the $XX_1\bar{1}$ for LH ($C_{3v}$, HH $h_2$) shows a splitting in the same oscillation phase as $XX_{lh}$, with FSS ~37 μeV in QD5. In the higher-energy side, $XX_{21}^+$ ($2e2h_11h_2$) exciton shows a clear polarization feature, well-depicted by the $C_{3v}$ transition scheme [30], which reflects the exchange $\Delta_{hh}$ between LH $h_1$ and HH $h_2$: ~150 μeV in QD1 and QD4 while ~205 μeV in QD2, QD5 and QD6 with more LH interlayer coupling. More LH coupling broadens the exciton linewidth in QD2 and populates a dominant $XX_{lh}$ in QD5 and QD6. In QD6, plenty of LH excitons in $C_{3v}$ spectral features are located in both sides of $XX_{lh}$ (FSS ~0) as mirror, e.g., $XX_{21}^+$ and $X\bar{1}_1^+$, $X_0\bar{1}$ and $XX_1\bar{1}$, $XX_2\bar{1}^+$ and $X_1\bar{1}^+$, reflecting higher symmetry; HH excitons X and XX are absent; the dominant $XX_{lh}$ and $X_{lh}$ show DLP ~30% related to VBM to HH $h_2$. $XX_{lh}$ usually shows FSS ~0 (QD1, QD3, QD4); in a charge field with LH interlayer coupling, it shows FSS ~12 μeV nearly constant (also for $X_{lh}$) (see QD2, QD5 and QD6), which reflects the eh exchange energy $\Delta^0_{eh}$ in the transition diagram [30]. In QD3 and QD4 with little coupling, the strain-induced LH–HH VBM shows the same DLP for X and XX, which is larger in QD3 with larger strain (DLP ~30%, FSS ~139 μeV). In QD1 and QD2 with $LH_x$ coupling and smaller FSS, XX shows larger DLP (30%) than X (DLP ~0). The negligible DLP in X is likely due to LH–HH VBM to the same $LH_X$ with more coupling. $XX_{lh}$ usually keeps the same DLP as XX (QD1, QD3, QD4). The LH delocalization usually reduces DLP. In QD2 (QD5) with more (less) LH coupling and smaller (larger) FSS, it shows smaller (larger) DLP than XX. The large DLP in $XX_{lh}$ in QD5 is consistent with the DLP in X, reflecting polarization-related LH tunneling for mixing.

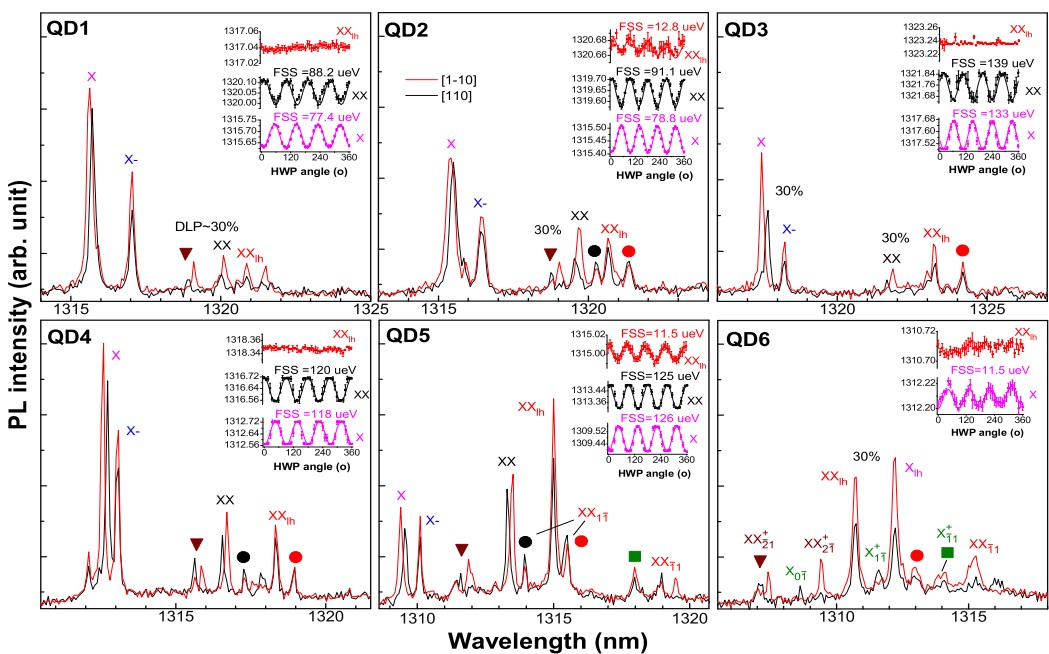

**Figure 3.** Polarization-resolved µPL spectra of QD1–6 with large FSS, black and red: orthogonal axes [110] and [1–10], large DLP ~30% marked. (**Insets**): FSS oscillations. HH excitons X and XX in large FSS and opposite signs; LH excitons marked, especially $XX_{lh}$ with small FSS, inverted triangle: $XX_{21}{}^+$, circles: $XX_{1\bar{1}}$ for both HH and LH, square: $X\bar{1}_1{}^+$, QD6: more LH excitons located around $XX_{lh}$.

### 3.4. QDs with LH Excitons in Small FSS

Since LHs have small effective mass and shallow barriers (see Figure 2), a little variation of QD dielectric environment (e.g., interface charges) will reduce their barriers for interlayer delocalization and show dominant LH excitons such as XX and $X_1\bar{1}{}^+$ or $XX_{21}{}^+$ with HH excitons absent, see Figure 4. Unlike the QDs in Figure 3, here, related to the environment charge field, the FSS of the LH excitons XX and X varies from <5 µeV to 22.6 µeV, and higher excitons are populated. In pump-power dependence, the near-1 slopes for all excitons reflect a fluent hole capture in delocalized LHs. The identification of the exciton complex is based on their similar spectral features in different QDs as referred to transition diagram [30], their mirrored positioning as referred to QD6 and their pump-power dependence slopes. Apart from the dominant $XX_{lh}$, biexcitons $XX_{21}{}^+$ and $XX_{1\bar{1}}$ usually show larger slopes than excitons X, $X_0\bar{1}$, $X_1\bar{1}{}^+$, $X^+$, $X^-$ and $X_0\bar{1}{}^-$. Under low pump, in QD7 there are X, $X_{01}$, $X^-$, $X_{01}{}^-$ in slope ~0.82 or 0.88, lower than $XX_{1\bar{1}}$ (~0.9); in QD8 there are $X\bar{1}_1{}^+$ and X in slopes ~0.93 or 0.74, lower than XX (~1.05) and $X^-$ (~0.84); in QD10 there are $XX_{21}{}^+$ and XX in slopes ~0.85, higher than X (~0.83) and $X\bar{1}_1{}^+$ (~0.82). Under high pump, X and $X\bar{1}_1{}^+$ are lost in the background. Excitons X, $X_0\bar{1}$, $X_1\bar{1}{}^+$ and $X_0\bar{1}{}^-$ keep nearly the same FSS oscillation phase opposite to biexcitons. In QD7 with many delocalized LHs, $XX_{lh}$ becomes like $X^-$ with an additional delocalized LH attracted by Coulomb interaction, similar for the other excitons: $XX_2\bar{1}{}^+$ becomes $XX_{1\bar{1}}$, $XX\bar{1}_1$ becomes $X_0\bar{1}{}^-$, $X_1\bar{1}{}^+$ becomes $X_0\bar{1}$. Meanwhile, the X is kept and a positive trion $X^+$ appears at the $X\bar{1}_1{}^+$ position with a large positive binding energy $E_B = V_{eh} - V_{hh}$ (i.e., $V_{hh} << V_{eh}$) [31,32]. For spatially delocalized LH levels, their Coulomb interaction ($V_{hh}$) is greatly reduced and the hole–hole exchange $\Delta_{hh}$ tends to be negligible. In this case, $XX_{21}{}^+$ and $X\bar{1}_1{}^+$ in QD9 and QD10 show FSS ~0; $XX_{1\bar{1}}$ in QD7 (QD10) shows the same FSS as X (XX), reflecting $\Delta^0{}_{eh}$ of 16~17 µeV. The $\Delta^0{}_{eh}$ as FSS of X and XX reflect is 14~22.6 µeV in the four QDs, a little larger than the $\Delta^0{}_{eh}$ of the QDs in Figure 3, ~12 µeV; they will vary greatly in a strong defect field as QD13~15, reflected in Figure 5. There is also a tiny X peak near the new $XX_{1\bar{1}}$, similar as QD8 and QD10. In QD10, different spectral features of $XX_{1\bar{1}}$ and $XX\bar{1}_1$ are clearly shown, as referred to $C_{3v}$ transition schemes [30]. In QD7 with $\Delta_{hh}$ ~0, the FSS ~40.5 µeV in $X_0\bar{1}$ reflects $\Delta^1{}_{eh}$ (e-h₂) [30]. In QD8 with less LH delocalization, apart from the dominant $XX_{lh}$ (FSS ~4.9 µeV with an additional

LH$_Z$ with FSS ~0), there is a dominant X$_{1\bar{1}}^+$ with FSS ~45.3 µeV and a secondary XX$_{1\bar{1}}$ with FSS ~56.3 µeV in opposite oscillations, reflecting a large Δ$_{hh}$ ~70 µeV with slight energy difference in Δ$_{eh}^0$ ~14.1 µeV (e-h$_1$) and Δ$_{eh}^I$ ~25 µeV (e-h$_2$) in C$_{3v}$ transition diagram [30]. QD7, QD9 and QD10 with delocalized LHs show population on XX$_{21}^+$ and other higher excitons located around XX$_{lh}$ (see QD9). In QD10, the hydrostatic charge field is high enough to vary the FSS oscillation phase and show a dominant XX$_{1\bar{1}}$ (i.e., X$^-$ coupled with an additional delocalized LH by Coulomb attraction) under high pump. In QD6~10 with LH delocalization, XX shows a negative E$_B$ (also X$^-$ in QD7, QD8, QD10, E$_B$ = V$_{eh}$ − V$_{ee}$) from smaller V$_{eh}$, V$_{eh}$ ≲ V$_{ee}$. In one sample, both QDs (large FSS and VBM or small FSS and LH exciton) will be found. To illustrate the small FSS in XX$_{lh}$, Figure 5 presents QD11~15. Unlike QD11 and QD12, QD13~15 with a hydrostatic charge field as screening show a weak exciton intensity with the FSS oscillation shift (lower-energy XX branch in [110]) and a clear spectral shoulder from the phonon. Around the minimum FSS ~0, the charge field varies the FSS oscillation phase quickly [33], e.g., 2.1 µeV in QD13. In QD15, the dominant XX with larger FSS shows oscillation shift clearly. For a more delocalized LH wavefunction, the tuning of FSS needs a higher charge field, and thus there are ~70% of QDs show XX$_{lh}$ FSS < 5 µeV as the statistics in Figure 5 reflect.

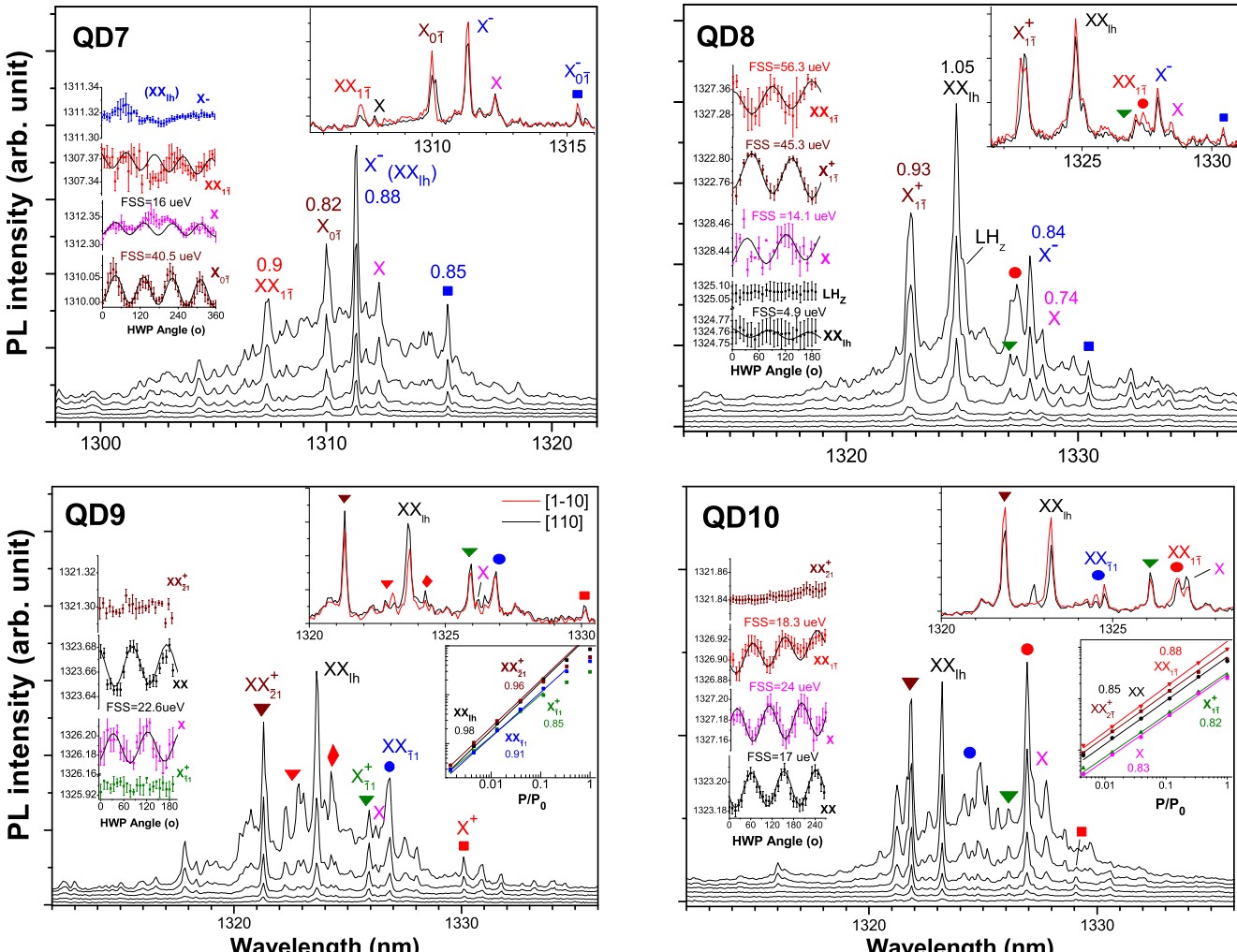

**Figure 4.** Pump-power-dependent µPL spectra of QD7–10 with LH excitons in small FSS. Inverted triangles: XX$_{21}^+$ (wine), X$\bar{1}_1^+$ (green), circles: XX$_{1\bar{1}}$ (red), XX$_{\bar{1}1}$ (blue), squares: X$^+$ (red), X$_0\bar{1}^-$ (blue), pink: X, black: XX$_{lh}$, red diamond in QD9: higher excitons. (**Insets: left**): FSS oscillations; (**top-right**): polarization-resolved spectra; (**bottom-right**): pump-power dependence, slope marked in plot or near peak.

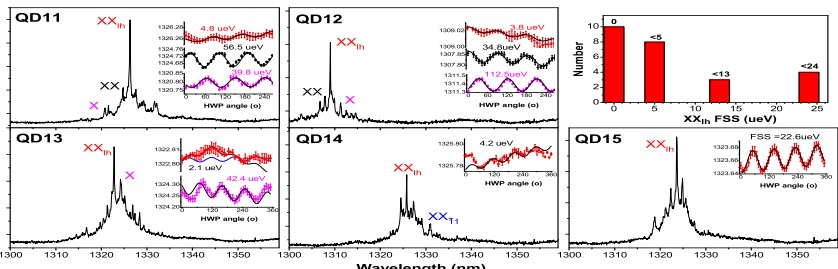

**Figure 5.** μPL spectra of QD11–15 with dominant $XX_{lh}$ in various FSS. QD13–15 with charge field to vary FSS oscillation phase and show phonon-related spectral shoulder. QD14 in similar spectrum as QD9. (**Top-right**): statistics on $XX_{lh}$ FSS as measured.

## 4. Conclusions

In this work, by microphotoluminescence (μPL) spectroscopy of bilayer single QDs, the strain effect on QD electronic structure, anisotropy, light hole (LH) formation and interlayer coupling have been uncovered from the spectral features, fine-structure splitting (FSS) and the degree of linear polarization. The biaxial strain enlarges QD anisotropy in [110] for large FSS ~100 μeV. It also induces LH excitons with small FSS (especially XX with FSS < 5 μeV, 70% of QDs). LH interlayer coupling greatly reduces the FSS of HH excitons with mixing and leads to dominant population on LH excitons XX, $XX_{11}$ and $XX_{21}^{+}$. If there are charge defects at the QD interface to build hydrostatic multi-electric fields, the exciton formation will be reduced and the emission from single QD will show an obvious spectral profile from the phonon scattering broadening; besides, the electric fields also vary the FSS oscillation phase. The μPL study combined with AFM morphology facilitates the optimization and the growth of the hybrid QD structure.

**Author Contributions:** X.S. (Xiangjun Shang), H.L., X.S. (Xiangbin Su), S.L., H.H. and D.D. took part in the optical measurement; H.L., X.S. (Xiangbin Su), Z.C. and H.N. took part in the sample growth; X.S. (Xiangjun Shang) and H.L. wrote the manuscript; Z.C., H.H., D.D. and H.N. participated in the discussions; H.N. and Z.N. supervised the writing of manuscript. All authors have read and agreed to the published version of the manuscript.

**Funding:** This research work is supported by the National Key Technologies R&D Program of China (Grant No. 2018YFA0306100), the Science and Technology Program of Guangzhou (Grant No. 202103030001), the Key-Area Research and Development Program of Guangdong Province (Grant No. 2018B030329001), the National Natural Science Foundation of China (Grant Nos. 62035017, 61505196), the Scientific Instrument Developing Project of Chinese Academy of Sciences (Grant No. YJKYYQ20170032), and the Program of Beijing Academy of Quantum Information Sciences (Grant No. Y18G01).

**Institutional Review Board Statement:** Not applicable.

**Informed Consent Statement:** Not applicable.

**Data Availability Statement:** The data that support the findings of this study are available from the corresponding author upon reasonable request.

**Conflicts of Interest:** The authors declare no conflict of interest.

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
