# Peer review of "Light Hole Excitons in Strain-Coupled Bilayer Quantum Dots with Small Fine-Structure Splitting"

_crystals, doi:10.3390/cryst12081116_

Round 1

Reviewer 1 Report

First of all, the explanations described in the text are difficult to follow for the reader and in my view most of them are speculative with no good arguments. However I find it interesting to the community to get acquainted with this work, so I recommend for publication after minor changes.

1) I would suggest rewritinig sections 3.2 and 3.3 to aviod long sentences which are split by “;” sign, e.g. page 7 line 222-225, which is:

In QD15, a dominant XX with spectral shoulder like QD13 and QD14 and larger FSS ~22.6 μeV (related to higher charge field) shows clearly the lower-energy XX branch along [110] V; the hydrostatic interface charge fields as screening could achieve FSS ~0 in QD exciton (e.g. 2.1 μeV in QD13).

The first part of this sentence is clear, however the second part about “hydrostatic interface charge fields” concerns which QD? Is is about QD 15 or QD 13? Do you mean that if the screening potential is present, the FSS is reduced? It is not clear.

I would ask similar question in a number of places, so please try to avoid separation like “;” and start another sentence which explains certain case.

2) As author say, the FSS of XX should be similar to X, however QD6 is obviously not the case. Concerning QD6 authors say that “original XX with large FSS becomes XX with FSS~0”. What do you mean? Are QD5 and QD6 somehow related to each other? It is not clear.

3) Authors obseved non-zero FSS for trions (QD2,QD5). Where it comes from? In my view, it comes from the experimental error, namely there exists other lines in close vicinity to X+ and their DLP could influence the fitting procedure, or it is simply not the trion, but some other QD exciton. How could you be sure that the spectra contains emission lines from exactly single QD?

4) Regarding Fig3. the XX intensity vs power dependence is linear for QD 9, 10 and slightly superlinear for QD 8, and the XX dependence should be quadratic, or at least faster compared to other basic excitonic states. I could not find good explanation in the text. Moreover, typically X or one of the trion state for low excitation power is observed, and for higher excitation power those lines saturate and they are lost in the background, which is not the case for any of the QDs. Could you explain that, as well?

Author Response

Dear reviewer,

We have revised the manuscript according to your comment. In the attachment we response the questions you ask. Thanks for your review on this work!

Reviewer 2 Report

To the best of my ability, I have read through and reviewed this manuscript, which describes micro-photoluminescence (PL) and atomic-force microscopy (AFM) studies of InAs/GaAs bilayer quantum dots in the presence of a strain-reducing layer. The authors see an enhanced anisotropic fine-structure splitting due to biaxial strain, but argue that a strain-induced light-hole interlayer coupling significantly reduces the fine-structure splitting, in particular for a biexciton line. The authors also argue that the combination of AFM and PL can be used to improve the design and fabrication of hybrid quantum-dot structures.

Overall, it appears that the measurements done were reasonable and could be publishable. However, I am not able to fully understand the measurements or the authors' interpretation of them because of a rather low-quality and confusing presentation. The main issue is the low quality of English in the manuscript. In some places I can look past this and interpret what the authors mean, but for most of the manuscript I am left guessing at the intended meaning. This is a problem throughout the entire manuscript; for me to be able to provide a reasonable review, the authors should first rewrite the document (probably with the help of a native English speaker).

In addition to problems with the English, I am unable to fully understand the figures from what information is given in the captions and in the text. I will give several examples:

-In the "Materials and Methods" section, it is very difficult to visualize the fabricated structure without a schematic showing the layers and their composition. There is a reference given (ref. [23]) for more details of the structure and growth, but that reference also does not have a schematic (there is an SEM image in [23], but no schematic). I was also surprised to see that Ref. [23] is from 2017. Are these the very same structures that the authors have studied for the past 5 years with no changes to the fabrication procedures?

-Figure 1: I understand from the caption and text that the blue curve in the top panel gives the PL for region 1, the blue curve in the bottom panel gives the PL for region 2, and the red curve in the bottom panel gives the PL for region 3, but what is the black curve in the top panel? It is not explained anywhere as far as I can tell.

-Figure 2: I find it confusing that the plots for QD1-6 are not shown sequentially. I don't understand why the FSS is shown for the X+ transition only for QD2 and QD5 (and not QD1,QD3,QD4,QD6). Is the claim that more lh coupling leads to a smaller FSS based *only* on a comparison of QD2 and QD5? I don't understand how this claim can be made with only two examples and when the FSS is so close (12.8 ueV vs 11.5 ueV). I wasn't able to find any statistical analysis of this effect in the paper -- it looks like a qualitative observation based on a single pair of dots?

-Figure 3: Again, the plots for QD8-10 are shown in a random order (the authors should reorder this unless there is a good reason for this order, in which case this should be explained clearly). I find it confusing to use the same color scheme on the PL curves for different powers as is used on the labels for the different peaks.

-Figure 4: Once again, the plots for QD11-15 are shown out of order. Why is this? I'm unable to understand the discussion of this figure related to LH coupling because of the confusing language used and grammatical inconsistencies.

In summary, I am unfortunately unable to provide a complete review of this manuscript. The English is too difficult for me to parse in many parts. In some places, where I think I can guess at the authors' intended meaning, I can see that parts of the figures are unexplained (or are not clearly explained) and I am unable to assess whether the observations support the authors' conclusions.

Author Response

(The authors gave the same response as above.)

Round 2

Reviewer 2 Report

In the revised manuscript, the authors have addressed some of my concerns associated with unclear explanations and labelling. My main concern about the quality of the English remains, but the remaining scientific content may be publishable with thorough copy editing. Some more detail follows:

-Inclusion of the new figure 2 is very helpful (this shows a schematic of the simulated band structure and bilayer QD).

-Inclusion of the new reference [26] (including a schematic of a related heterostructure) is also very helpful.

-The labelling in Figs. 3,4,5, has been greatly improved and the inclusion of the upper-right inset of Fig. 5 showing statistics of the FSS was essential to establish the claimed trend of lower FSS for LH. This is much appreciated.

Overall, I believe the data and resulting analysis are sound and publishable. I do hope the authors can take the time to further improve the English in the manuscript. The quality of the English is similar to what comes out of a bad translation engine. Often words are used incorrectly in a way that can lead to confusion or misinterpretation. This is problematic throughout the manuscript (not limited to the example below), but I will indicate specifically where this causes problems in an example to make the point clear:  

As an example, consider the text on p. 3:
“Single QDs can be filtered (spatially isolated) by DBR cavity (e.g. pillar) [27,30], in addition to well control of indium coverage and strain site (see the spectrum in black curve, i.e. another point in region 1, with individual QDs at λ ~1.3 μm in sharp exciton lines). It is unlike single QDs at λ <1.1 μm with 2nd critical coverage where a broad T exists (both region 1 and region 2 show exciton lines in the spectral profile)”

The grammar and syntax of the first sentence is not very good, but I can roughly understand the intended meaning. In the second sentence, “It is unlike single QDs...”, I am not sure what is meant here by “It”. Is the intention to say that the spectrum is not consistent with single quantum dots in that wavelength range? I’m not sure. Similarly, what is meant by “...with 2nd critical coverage where a broad T exists...”. I presume “T” here is the temperature during the growth process? But what does it mean for a ‘broad’ T to ‘exist’ with the 2nd critical coverage? Does this mean that the growth was performed without fine control over T, or that there was a variation of temperature during growth? I really cannot understand.

I hope the authors can take the time to clarify the English in their paper. If they can do this, I believe the data and scientific findings are publishable.

Author Response

Dear reviewer,

We have revised the manuscript according to your suggestions and response to the questions you asked. Kindly check them in the attached file.
